

# Quality assessment from biobank plasma and serum specimens: a systematic review

Xiongshun Liang[1,2], Wanna Xu[1,2], Lin Chen[1,2], Xiaoqian Liu[1,2], Wenxu Hong[1,2] and Xuqiao Hu[1,2]

[1] Shenzhen Center for Chronic Disease Control, Shenzhen, China
[2] Shenzhen Institute of Dermatology, Shenzhen, China

## ABSTRACT

Accurate assessment of serum and plasma quality is essential for reliable biobanking and research. This systematic review synthesizes 46 studies and shows that current evaluation methods prioritize sample processing delays, freeze-thaw cycles, storage duration, and temperature variations, while neglecting preanalytical factors like medication and diet. Temperature critically affected stability: non-refrigerated samples (8–37 °C) showed 3.2 times more biomarker alterations than refrigerated samples after >24 h delays. Freeze-thaw cycles caused dose-dependent instability, with >10 cycles triggering severe degradation (70% altered biomarkers) and universal enzyme impairment. Even ≤5 cycles significantly altered enzymes (43% affected). Enzymes also degraded over time at <−20 °C, with alteration rates rising from 20% (1–5 years) to 55% (>10 years). The analysis consolidates stability data for enzymes, sterols, cytokines, and metabolites across conditions, providing an evidence-based foundation for quality control. Standardizing procedures to address these gaps will enhance sample integrity, research reliability, and clinical utility. Critically, this review underscores that preexisting biobank collections may have inherent limitations, necessitating careful evaluation of sample history when designing studies.

# INTRODUCTION

## Current status of serum and plasma quality assessment in biobanks

Biobanks are vital infrastructure for modern biomedical research, advancing disease mechanism understanding, genomics, and precision medicine implementation (*Gallagher, Ginsburg & Musick, 2024*; *Kiseleva et al., 2021*; *Perry et al., 2019*). Despite their critical importance, the increasing complexity of sample collection, storage, and processing has introduced significant challenges in maintaining the stability, consistency, and reliability of sample quality. These challenges profoundly impact the reproducibility and validity of research findings (*Dagher, 2022*). For instance, the US biobanks collectively hold over 300 million samples,, with an annual growth rate of 20 million. However, as the volume of collected samples increases, issues related to sample quality have become increasingly evident. Surveys reveal that 47% of researchers encounter difficulties in obtaining samples that meet the quality standards required for their studies. Additionally, 60% of researchers express concerns regarding the accuracy of their results, often attributing these concerns to

Corresponding authors
Wenxu Hong,
szbloodcenter@hotmail.com
Xuqiao Hu, haniahu@hotmail.com

sample quality issues. Moreover, 81% report that their research has been constrained by insufficient access to high-quality samples (*Baker, 2012*).

Inaccurate results lack scientific validity, misallocate resources, and distort research/clinical decisions. In the US, >30% of irreproducible biological findings stem from preanalytical errors, costing >$8 billion annually (*de Gramont et al., 2015*). This underscores a critical paradox: biobank expansion coexists with persistent sample quality challenges, severely impacting scientific efficiency. Thus, enhancing sample quality, optimizing management, and maximizing resource utility are urgent priorities.

To address these challenges, international organizations have established best practices to support quality control (QC) and quality assurance (QA) within biobanks (*Barnes et al., 2008*; *Garcia et al., 2014*; *Vaught et al., 2012*). Simultaneously, the Biobank Scientific Working Group has developed standardized preanalytical codes (SPREC) to ensure the proper handling of bio-samples during analysis (*Betsou et al., 2010*). These initiatives optimize high-quality sample collection, storage, and distribution, minimize preanalytical variable impacts, and enhance analytical accuracy. However, current practices focus on operational recommendations and lack application-specific guidance. Clinically, uncontrollable external factors further compound preanalytical effects (*Agrawal et al., 2018*).

## Research questions

Despite progress made in sample quality assessment, systematic reviews focusing on the preanalytical factors affecting sample quality remain limited. Meanwhile, with the increasing application of highly sensitive technologies such as mass spectrometry and single-cell sequencing in biomedical research, the standards for sample integrity have become significantly more stringent (*Nicolardi et al., 2010*; *Tang et al., 2019*). Consequently, there is an urgent need for researchers to establish a scientific, comprehensive, and practical framework for sample quality evaluation.

In response to this need, the present study aims to address the following research questions through a systematic review, with the goal of providing a reference for the quality management of serum and plasma samples:

1. What are the current trends in research on serum and plasma sample quality?
2. Which factors significantly influence the quality of serum and plasma samples?
3. What biomarkers effectively reflect the quality of serum and plasma samples?
4. What methods and technologies are currently employed to evaluate the quality of serum and plasma samples?

By answering these questions, this study seeks to integrate the latest findings in the field, propose recommendations for optimizing the quality management of biological samples, and provide theoretical foundations and practical tools for quantitative evaluation tailored to the needs of different types of research.

**Table 1 Inclusion and exclusion criteria.**

| Inclusion criteria | Exclusion criteria |
|---|---|
| At least one of the outcome measures of interest reported in the study orcalculable from the published data | Animal or cell line studies |
| Empirical studies | Biobanked serum or plasma samples other than such as tissue, urine and saliva |
| Written in English | Use of pre-servatives during collection, processing and storage other than standard |
| | Outcomes of interest that were not reported or could not be calculated from the original published data |
| | Literature reviews, commentaries or meta-analysis |
| | Written in other languages |

## MATERIALS AND METHODS

### Data collection and processing

Following established methodological standards (*Knobloch, Yoon & Vogt, 2011*), we conducted a systematic literature search in PubMed and Web of Science in September 2024, with the application of Boolean logic (("Biobank") AND ((("Quality Assessment") OR ("Quality Control")) OR ("stability"))) AND (("serum") OR ("plasma")). The search strategy was independently performed by two authors (Xiongshun-Liang and Wanna-Xu). In case of any discrepancies, the final decision was made by author Chen Lin. In order to guarantee both the quality and reliability of the sources, only peer-reviewed journal articles with full text access were considered for inclusion. The article defines the inclusion and exclusion criteria outlined in Table 1, and each study was thoroughly reviewed to assess its eligibility for analysis.

### Data extraction

Data extraction was performed independently by two authors (Xiongshun-Liang and Wanna-Xu). The following data were extracted from eligible studies: study characteristics, including first author, country, year of publication and sample size; diagnostic quality measures, including time to delayed separation, delayed separation temperature, number of freeze-thaw cycles, storage time and storage temperature and predictive quality measures, including group, measurand, stability, measurement method and types of measurand. Any disagreements in the data extraction process were addressed through discussion between the two authors. For studies where the required data was not available in the full text, the corresponding authors were not approached.

### Statistical analysis

The systematic review was performed and reported in accord-ance with the PRISMA guidelines for systematic reviews (*Page et al., 2021*). The data analysis for this study was generated using the Microsoft Excel 2016.

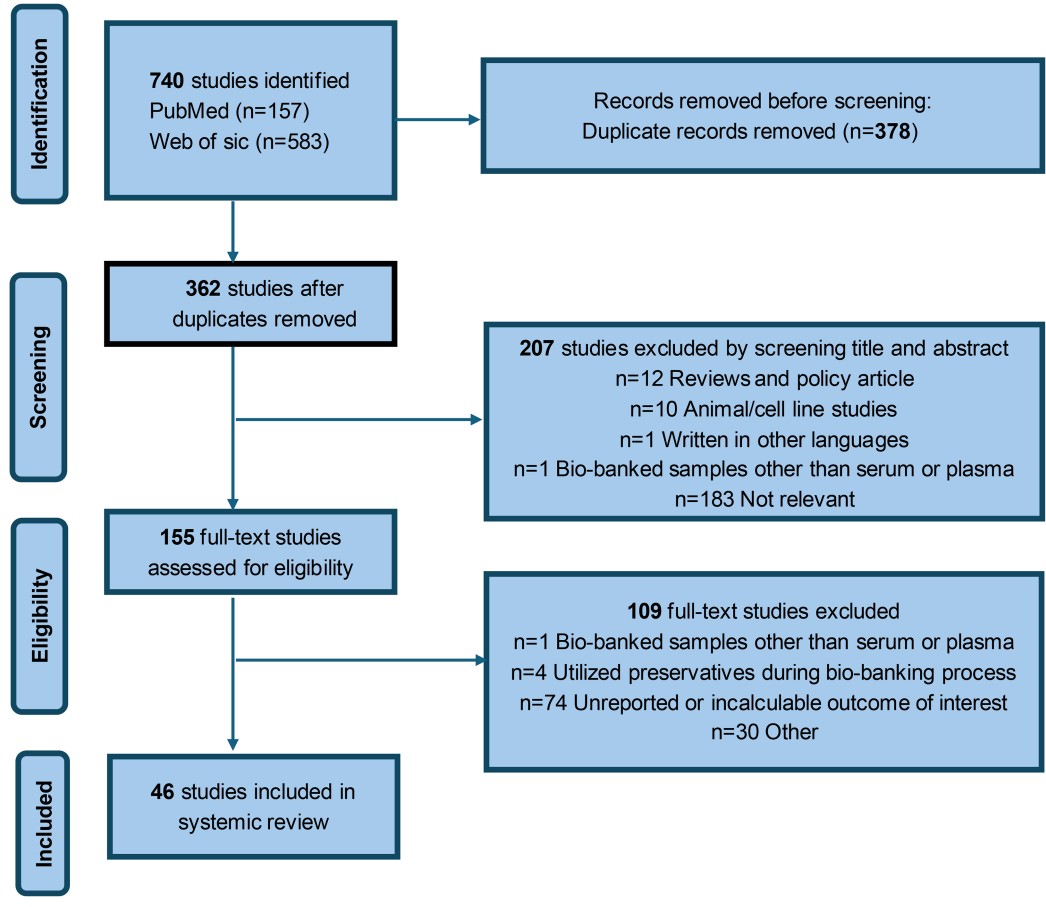

**Figure 1 Flow chart for study identification and inclusion.**

## RESULTS

### Summary of eligible studies

Figure 1 presents a flowchart detailing the literature search process. The initial search of relevant databases with predefined keywords identified 362 studies. After evaluating the titles and abstracts, 155 studies were considered potentially relevant for further review. However, after a full-text assessment, 109 studies were excluded for various reasons: one study involved biobank samples other than serum or plasma, four studies used preservatives during the biobank processing, and 74 studies either failed to report the desired outcomes or presented data that could not be analyzed. Consequently, 46 studies satisfied the inclusion criteria and were included in the review. Supplemental Material Table S1 summarizes the key findings from all eligible studies. Specifically, 19 studies reported the impact of delayed processing on sample quality, 15 studies documented the effects of methodological freeze-thaw cycles on sample quality, and 26 studies investigated the influence of cryopreservation duration and temperature on sample quality (*Alegre et al., 2022*; *Araujo et al., 2018*; *Barroso, Farinha & Guimarães, 2018*; *Chaigneau et al.,*

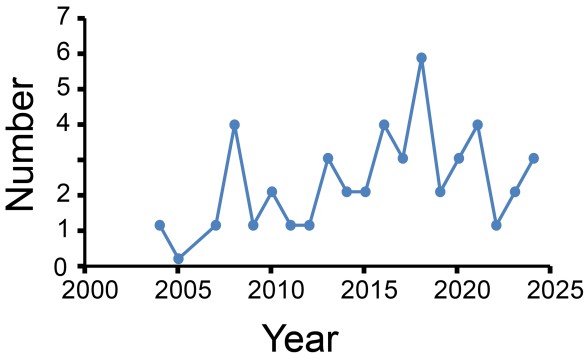

**Figure 2 Number of published articles per year.**

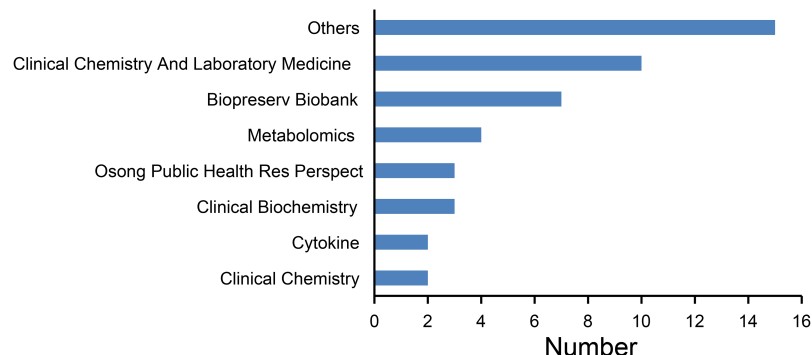

**Figure 3 Number of articles published by each journal.**

*2007*; *Cuhadar et al., 2013*; *Dard et al., 2017*; *Dorow et al., 2016*; *Freiburghaus et al., 2020*; *Gao et al., 2021*; *Gislefoss, Grimsrud & Morkrid, 2009*; *Gislefoss, Grimsrud & Mørkrid, 2008*, *2015*; *Gislefoss et al., 2017*; *Hannisdal et al., 2010*; *Hernestal-Boman et al., 2010*; *Huh et al., 2023*; *Jensen et al., 2023*; *Jiang et al., 2020*; *Kamlage et al., 2014*, *2018*; *Kang et al., 2013*; *Kim et al., 2012*; *Kisand et al., 2011*; *Kofanova et al., 2018*, *2024*; *La Frano et al., 2018*; *Lee et al., 2016a*; *Lee, Kim & Shin, 2015*; *Lee et al., 2016b*; *Lengelle, Panopoulos & Betsou, 2008*; *Mondésert et al., 2024*; *Paltiel et al., 2008*; *Pan et al., 2020*; *Rolandsson et al., 2004*; *Schwarz et al., 2019*; *Shen et al., 2018*; *Skogstrand et al., 2008*; *Späth et al., 2021*; *Trezzi et al., 2016*; *van Waateringe et al., 2017*; *Verberk et al., 2021*; *Vincent et al., 2019*; *Wang et al., 2024*; *Yin et al., 2013*; *Zander et al., 2014*; *Zheng et al., 2021*).

## Research trends

Figure 2 shows annual publication outputs from 2004 to 2024. Peak publication occurred in 2018 ($n = 6$), with secondary peaks in 2008, 2016, and 2021 (each $n = 4$), with no more than four articles every year thereafter. However, the trend shows a gradual decline in research enthusiasm for this important field (Fig. 2). This also reflects researchers' insufficient understanding of biological sample quality control.

The journal distribution is shown in Fig. 3. The most published journals were *Clinical Chemistry and Laboratory Medicine* (10), *Biopreservation Biobanking* (seven) and

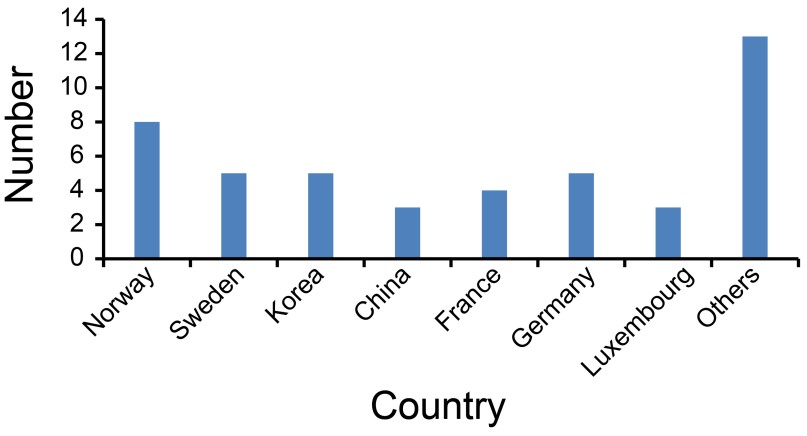

**Figure 4  Number of articles published by each country.**

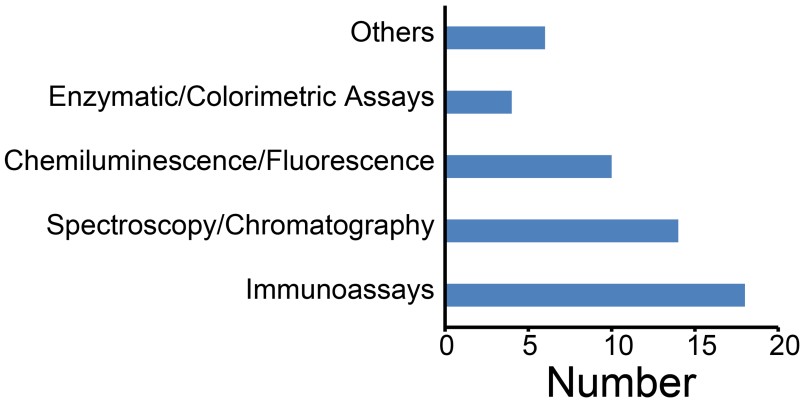

**Figure 5  Techniques for assessing the quality of serum and plasma samples.**

*Metabolomics* (four). Other less frequently published journals include *Osong Public Health Res Perspect*, *Clinical Chemistry*, *Cytokine*, *Thrombosis Research*.

The country distribution is shown in Fig. 4. The most publication countries were Norway (eight), Sweden (five), Germany (five), Korea (five) and France (four). Other less frequently published countries include China, Luxembourg, Netherlands.

Through analysis of the included literature, we systematically reviewed the application of measurement methods for assessing serum and plasma sample qualities. Immunoassays predominated, notably enzyme-linked immunosorbent assays (ELISA), followed by spectroscopy/chromatography techniques—including Liquid Chromatography-Tandem Mass Spectrometry (LC-MS/MS), Gas Chromatography-Mass Spectrometry (GC-MS), Nuclear Magnetic Resonance Spectroscopy (NMR) spectroscopy (Fig. 5). Immunoassays are primarily applied to detect specific biomarkers such as cytokines and antibodies. Whereas spectroscopy/chromatography is recognized for its high precision and sensitivity in quantifying metabolites, proteins, and other biomarkers, chemiluminescence assays are

**Table 2  Impact of pre-analytical delays on biomarker stability.**

| Delayed separation temperature (°C) | Delayed separation (h) | Biomarker ontology class (Number) | Significantly changed biomarkers *vs.* the control group (Increase/Decrease/Unclear[a]) | |
|---|---|---|---|---|
| | | | **Number** | **Percent change** |
| T ≤ 8 | H ≤ 4 | ALL (449) | 30 (25/5/0) | 6.7 (5.6/1.1/0.0) |
| | | Metabolites (243) | 7 (4/3/0) | 2.9 (1.6/1.2/0.0) |
| | | Enzymes (7) | 6 (5/1/0) | 85.7 (71.4/14.3/0.0) |
| | | Cytokines (55) | 7 (7/0/0) | 12.7 (12.7/0.0/0.0) |
| | | Sterols (1) | 1 (1/0/0) | 100.0 (100.0/0.0/0.0) |
| | | Other (143) | 9 (8/1/0) | 6.3 (5.6/0.7/0.0) |
| | 4 < H ≤ 24 | ALL (185) | 24 (21/3/0) | 13.0 (11.4/1.6/0.0) |
| | | Metabolites (3) | 0 (0/0/0) | 0.0 (0.0/0.0/0.0) |
| | | Enzymes (7) | 3 (2/1/0) | 42.9 (28.6/14.3/0.0) |
| | | Cytokines (33) | 6 (5/1/0) | 18.2 (15.2/3.0/0.0) |
| | | Sterols (1) | 0 (0/0/0) | 0.0 (0.0/0.0/0.0) |
| | | Other (141) | 15 (14/1/0) | 10.6 (9.9/0.7/0.0) |
| | H > 24 | ALL (167) | 28 (28/0/0) | 16.8 (16.8/0.0/0.0) |
| | | Metabolites (0) | 0 (0/0/0) | 0.0 (0.0/0.0/0.0) |
| | | Enzymes (1) | 0 (0/0/0) | 0.0 (0.0/0.0/0.0) |
| | | Cytokines (27) | 7 (7/0/0) | 25.9 (25.9/0.0/0.0) |
| | | Sterols (0) | 0 (0/0/0) | 0.0 (0.0/0.0/0.0) |
| | | Other (139) | 21 (21/0/0) | 15.1 (15.1/0.0/0.0) |
| 8 < T ≤ 37 | H ≤ 4 | ALL (441) | 91 (39/4/48) | 20.6 (8.8/0.9/10.9) |
| | | Metabolites (247) | 55 (6/1/48) | 22.3 (2.4/0.4/19.4) |
| | | Enzymes (3) | 1 (1/0/0) | 33.3 (33.3/0.0/0.0) |
| | | Cytokines (42) | 19 (17/2/0) | 45.2 (40.5/4.8/0.0) |
| | | Sterols (0) | 0 (0/0/0) | 0.0 (0.0/0.0/0.0) |
| | | Other (149) | 16 (15/1/0) | 10.7 (10.1/0.7/0.0) |
| | 4 < H ≤ 24 | ALL (465) | 159 (134/25/0) | 34.2 (28.8/5.4/0.0) |
| | | Metabolites (267) | 83 (66/17/0) | 31.1 (24.7/6.4/0.0) |
| | | Enzymes (3) | 2 (1/1/0) | 66.7 (33.3/33.3/0.0) |
| | | Cytokines (46) | 31 (26/5/0) | 67.4 (56.5/10.9/0.0) |
| | | Sterols (0) | 0 (0/0/0) | 0.0 (0.0/0.0/0.0) |
| | | Other (149) | 43 (41/2/0) | 28.9 (27.5/1.3/0.0) |
| | H > 24 | ALL (618) | 334 (70/4/260) | 54.0 (11.3/0.6/42.1) |
| | | Metabolites (428) | 260 (0/0/260) | 60.7 (0.0/0.0/60.7) |
| | | Enzymes (2) | 2 (2/0/0) | 100.0 (100.0/0.0/0.0) |
| | | Cytokines (41) | 27 (25/2/0) | 65.9 (61.0/4.9/0.0) |
| | | Sterols (0) | 0 (0/0/0) | 0.0 (0.0/0.0/0.0) |
| | | Other (147) | 45 (43/2/0) | 30.6 (29.3/1.4/0.0) |

**Note:**
[a] The literature reports statistically significant alterations but fails to provide numerical values specifying the magnitude of increases or decreases.

commonly employed for electrolyte analysis. Consequently, methodological selection depends on specific research objectives and sample characteristics.

## Factors affecting the quality of serum and plasma samples
### Impact of preanalytical delays on biomarker stability

Temperature is the key factor affecting biomarker stability during preanalytical delays (Table 2). Refrigerated samples (≤8 °C) remain highly stable, with overall changes staying below 16.8% even after 24-h delays (*Jensen et al., 2023*; *Shen et al., 2018*; *Skogstrand et al., 2008*). In contrast, non-refrigerated samples (8–37 °C) degrade significantly beyond 24 h, showing a 54.0% alteration rate—3.2 times higher than refrigerated samples (*Jensen et al., 2023*; *Shen et al., 2018*; *Skogstrand et al., 2008*). This strong temperature dependence highlights the essential need for continuous refrigeration, especially since metabolites degrade rapidly at room temperature (60.7% altered beyond 24 h) (*La Frano et al., 2018*).

Critical time thresholds reveal biomarker-specific vulnerabilities. Enzymatic activity in refrigerated blood is highly unstable: 85.7% of enzymes change significantly within just 4 h, primarily showing increased activity (71.4%), suggesting catalysis accelerates even when cold. After 24 h at room temperature, metabolites show widespread changes (60.7%) and lose all directional consistency (100% indeterminate changes), indicating metabolic network disruption (*Kang et al., 2013*; *Lee et al., 2016b*). Cytokines are sensitive to both time and temperature: alterations increase with delay time and are amplified without refrigeration (45.2–67.4%). Notably, over 90% of cytokine changes involve concentration increases, likely due to sustained release from blood cells during delays (*Gao et al., 2021*; *Kofanova et al., 2018*; *Pan et al., 2020*; *Skogstrand et al., 2008*; *Verberk et al., 2021*).

These findings establish practical priorities for sample handling. Enzymes require processing within 4 h, regardless of refrigeration. Metabolomics analysis becomes invalid beyond 24 h at room temperature due to irreversible degradation. Cytokine measurement needs strict temperature control and processing within 4 h to prevent artificial elevation.

### Impact of freeze-thaw cycles on biomarker stability

Our systematic analysis reveals a dose-dependent relationship between freeze-thaw cycles (N) and biomarker instability in serum/plasma samples, with significant class-specific variations (Table 3). Samples subjected to >10 cycles exhibited profound instability, with 70.0% (7/10) of all biomarkers significantly altered, predominantly reductions (50.0%) (*Chaigneau et al., 2007*; *Paltiel et al., 2008*).

In the moderate range (5 < N ≤ 10 cycles), overall instability dropped substantially to 10.2% (15/147). However, metabolites and enzymes remained highly vulnerable, each exhibiting a 33.3% (5/15) alteration rate, featuring a mix of increases and decreases (20.0% increase, 13.3% decrease for both classes). Cytokines showed a lower alteration rate (20.0%, 1/5, decrease), while sterols remained stable (0/6) (*Chaigneau et al., 2007*; *Cuhadar et al., 2013*; *Gao et al., 2021*; *Gislefoss et al., 2017*; *Kang et al., 2013*; *Paltiel et al., 2008*; *Shen et al., 2018*).

Critically, even minimal freeze-thaw exposure (N ≤ 5 cycles) induced alterations in 15.3% (97/633) of all biomarkers. Metabolites exhibited a 16.7% (81/485) alteration rate,

**Table 3 Impact of freeze-thaw cycles on biomarker stability.**

| Freeze/thaw cycles (Number) | Biomarker ontology class (Number) | Significantly changed biomarkers *vs.* the control group (Increase/Decrease/Unclear)[a] | |
|---|---|---|---|
| | | **Number** | **Percent change** |
| N ≤ 5 | ALL (633) | 97 (18/11/68) | 15.3 (2.8/1.7/10.7) |
| | Metabolites (485) | 81 (8/5/68) | 16.7 (1.6/1.0/14) |
| | Enzymes (14) | 6 (3/3/0) | 42.9 (21.4/21.4/0) |
| | Cytokines (22) | 6 (5/1/0) | 27.3 (22.7/4.5/0) |
| | Sterols (5) | 1 (0/1/0) | 20.0 (0.0/20.0/0) |
| | Other (107) | 3 (2/1/0) | 2.8 (1.9/0.9/0) |
| 5 < N ≤ 10 | ALL (147) | 15 (9/6/0) | 10.2 (6.1/4.1/0) |
| | Metabolites (15) | 5 (3/2/0) | 33.3 (20.0/13.3/0) |
| | Enzymes (15) | 5 (3/2/0) | 33.3 (20.0/13.3/0) |
| | Cytokines (5) | 1 (0/1/0) | 20.0 (0.0/20.0/0) |
| | Sterols (6) | 0 (0/0/0) | 0.0 (0.0/0.0/0) |
| | Other (106) | 4 (3/1/0) | 3.8 (2.8/0.9/0) |
| N > 10 | ALL (10) | 7 (2/5/0) | 70.0 (20.0/50.0/0) |
| | Metabolites (2) | 2 (1/1/0) | 100.0 (50.0/50.0/0) |
| | Enzymes (3) | 3 (1/2/0) | 100.0 (33.3/66.7/0) |
| | Cytokines (1) | 0 (0/0/0) | 0.0 (0.0/0.0/0) |
| | Sterols (1) | 1 (0/1/0) | 100.0 (0.0/100.0/0) |
| | Other (3) | 1 (0/1/0) | 33.3 (0.0/33.3/0) |

**Note:**
[a] The literature reports statistically significant alterations but fails to provide numerical values specifying the magnitude of increases or decreases.

but demonstrated a complex response pattern distinct from other classes. In stark contrast, enzymes were the most sensitive class at low cycles, exhibiting the highest relative alteration rate (42.9%, 6/14) with clear directional shifts (21.4% increase, 21.4% decrease). Cytokines showed moderate instability (27.3%, 6/22), primarily increases (22.7%) (*Araujo et al., 2018*; *Cuhadar et al., 2013*; *Dorow et al., 2016*; *Gislefoss et al., 2017*; *Kang et al., 2013*; *La Frano et al., 2018*; *Paltiel et al., 2008*; *Yin et al., 2013*).

### Impact of storage duration and temperature on biomarker stability

Extended cryopreservation (>10 years at T < −20 °C) caused the most severe biomarker alterations (45.8%, 27/59). Metabolites showed highest instability (83.3%, 5/6; predominantly decreases: 66.7%), followed by enzymes (54.5%, 6/11; mixed changes). Sterols exhibited universal alteration (100%, 3/3) despite limited samples, while >50% of "Other" biomarkers changed (52.6%, 10/19; mostly increases: 42.1%) (*Alegre et al., 2022*; *Gislefoss, Grimsrud & Morkrid, 2009*, *2008*; *Gislefoss, Grimsrud & Mørkrid, 2015*; *Rolandsson et al., 2004*). These results demonstrate significant degradation of critical biomarkers beyond 10 years, even at ultra-low temperatures.

Shorter cryopreservation (1–5 years at T < −20 °C) yielded lower overall alterations (30.8%, 45/146), but class-specific vulnerabilities persisted: sterols (71.4%, 5/7 (*Gislefoss, Grimsrud & Morkrid, 2009*; *Paltiel et al., 2008*; *Wang et al., 2024*), metabolites (41.2%,

**Table 4 Impact of storage duration and temperature on biomarker stability.**

| Storage temperature (°C) | Storage time (years) | Biomarker ontology class (Number) | Significantly changed biomarkers vs. the control group (Increase/Decrease)[a] | |
|---|---|---|---|---|
| | | | Number | Percent change |
| T < −20 | 1–5 | ALL (146) | 45 (10/35) | 30.8 (6.8/24.0) |
| | | Metabolites (51) | 21 (2/19) | 41.2 (3.9/37.3) |
| | | Enzymes (15) | 3 (0/3) | 20.0 (0.0/20.0) |
| | | Cytokines (28) | 4 (3/1) | 14.3 (10.7/3.6) |
| | | Sterols (7) | 5 (2/3) | 71.4 (28.6/42.9) |
| | | Other (45) | 12 (3/9) | 26.7 (6.7/20.0) |
| | 5–10 | ALL (22) | 3 (0/3) | 13.6 (0.0/13.6) |
| | | Metabolites (0) | 0 (0/0) | 0.0 (0.0/0.0) |
| | | Enzymes (3) | 0 (0/0) | 0.0 (0.0/0.0) |
| | | Cytokines (18) | 3 (0/3) | 16.7 (0.0/16.7) |
| | | Sterols (0) | 0 (0/0) | 0.0 (0.0/0.0) |
| | | Other (1) | 0 (0/0) | 0.0 (0.0/0.0) |
| | >10 | ALL (59) | 27 (12/15) | 45.8 (20.3/25.4) |
| | | Metabolites (6) | 5 (1/4) | 83.3 (16.7/66.7) |
| | | Enzymes (11) | 6 (2/4) | 54.5 (18.2/36.4) |
| | | Cytokines (20) | 3 (0/3) | 15.0 (0.0/15.0) |
| | | Sterols (3) | 3 (1/2) | 100.0 (33.3/66.7) |
| | | Other (19) | 10 (8/2) | 52.6 (42.1/10.5) |
| −20 ≤ T | 1–5 | ALL (47) | 18 (6/12) | 38.3 (12.8/25.5) |
| | | Metabolites (9) | 3 (2/1) | 33.3 (22.2/11.1) |
| | | Enzymes (11) | 4 (0/4) | 36.4 (0.0/36.4) |
| | | Cytokines (2) | 1 (0/1) | 50.0 (0.0/50.0) |
| | | Sterols (2) | 0 (0/0) | 0.0 (0.0/0.0) |
| | | Other (23) | 10 (4/6) | 43.5 (17.4/26.1) |
| | 5–10 | ALL (44) | 18 (18/0) | 40.9 (40.9/0.0) |
| | | Metabolites (34) | 17 (17/0) | 50.0 (50.0/0.0) |
| | | Enzymes (0) | 0 (0/0) | 0.0 (0.0/0.0) |
| | | Cytokines (7) | 1 (1/0) | 14.3 (14.3/0.0) |
| | | Sterols (0) | 0 (0/0) | 0.0 (0.0/0.0) |
| | | Other (3) | 0 (0/0) | 0.0 (0.0/0.0) |

**Note:**

[a] The literature reports statistically significant alterations but fails to provide numerical values specifying the magnitude of increases or decreases.

21/51; mainly decreases (*Araujo et al., 2018*; *Dorow et al., 2016*; *Freiburghaus et al., 2020*; *Gislefoss, Grimsrud & Morkrid, 2009*, *2008*; *Hannisdal et al., 2010*; *Mondésert et al., 2024*; *Paltiel et al., 2008*; *Wang et al., 2024*; *Zander et al., 2014*). Enzymes and cytokines remained more stable (20.0% and 14.3%). At −20 °C ≤ T (1–5 years), alterations reached 38.3% (18/47).

Intermediate storage (5–10 years at −20 ≤ T) revealed exclusive increases (40.9%, 18/44), primarily metabolite-driven (50.0%, 17/34) (*Dorow et al., 2016*). This

contrasts sharply with metabolite decreases observed during long-term ultra-low temperature storage. Enzymes showed progressive degradation at T < −20 °C: alterations increased from 20.0% (3/15; all decreases) at 1–5 years to 54.5% (6/11) at >10 years (*Alegre et al., 2022*; *Freiburghaus et al., 2020*; *Gislefoss, Grimsrud & Morkrid, 2009, 2015*; *Hernestal-Boman et al., 2010*; *Kofanova et al., 2024*; *Mondésert et al., 2024*; *Paltiel et al., 2008*; *Zander et al., 2014*), confirming time-dependent instability (Table 4).

### Other factors

The choice of blood collection tubes has a significant impact on the quality of serum and plasma samples. Studies have shown that blood collection tubes containing sodium fluoride effectively preserve the integrity of blood metabolites, particularly in cases of delayed sample processing, by substantially reducing fluctuations in glucose and lactate levels (*Xiong et al., 2024*). Furthermore, different anticoagulants, such as EDTA and lithium heparin, exhibit varying effects on cytokines and biochemical indicators. For instance, EDTA plasma samples demonstrate higher cytokine stability during handling and processing (*Verberk et al., 2021*). Hemolysis is another critical factor affecting the quality of serum and plasma samples. It not only interferes with biochemical analyses but also compromises the stability and reliability of certain biomarkers (*Rolandsson et al., 2004*; *Yin et al., 2013*).

# DISCUSSION

## Research trends and standardization challenges in quality control of serum and plasma samples

The quality assessment of serum and plasma samples is pivotal in ensuring the reliability of research outcomes within biobank studies and their subsequent applications. Modern pathology and molecular analyses are fundamentally reliant on high-quality human bio-specimens, as compromised samples can substantially undermine the accuracy of these analyses. The publication of unreliable data may, in turn, reduce the reproducibility of biomedical research and hinder clinical translation outcomes (*Betsou et al., 2013*). Research on the quality assessment of serum and plasma samples prior to analysis has not shown a consistent upward trend in recent years. The number of studies has fluctuated over time, with a notable increase between 2014 and 2018. This rise is likely linked to advancements in biomedical research, the growth of biobanks, and the increasing demand for high-quality biological samples. In contrast, the past 5 years have seen a relative decline in publications on this topic. This may reflect a lack of sufficient attention to the quality assessment of samples before analysis. For instance, among the 46 articles published in the last 2 years, only six addressed this issue. Notably, 42% of these articles focused on delayed processing, 33% examined the effects of repeated freezing and thawing, and 57% investigated the influence of storage temperature and duration. However, the literature remains sparse concerning the effects of factors such as medication use, dietary habits, and genetic history prior to sample collection.

Pre-collection variables such as medication use, dietary intake, and circadian rhythms significantly influence the integrity of serum and plasma biomarkers but are frequently

neglected in biobanking protocols. Pharmacological agents like NSAIDs can suppress inflammatory markers such as IL-6 and CRP, while anticoagulants may interfere with assay accuracy through matrix effects (*Akácsos-Szász et al., 2024*; *Ruhnau et al., 2023*). Inconsistent fasting protocols, especially non-standardized lipid profile collection, may alter metabolite measurements such as apolipoproteins (*e.g.*, ApoC-I and ApoA-II) and glucose levels, as highlighted in studies requiring strict overnight fasting for pancreatic biomarker analysis (*Xue et al., 2010*). Concurrently, sleep deprivation-induced elevations in cytokines (*e.g.*, IL-1β) and lifestyle factors (*e.g.*, physical activity) further underscore the need for detailed pre-collection documentation (*Ng et al., 2024*). Incorporating pre-collection questionnaires and computational adjustments for these variables into biobank workflows will improve data reliability and facilitate cross-study comparisons.

Our findings underscore the critical importance of temperature-controlled and time-sensitive handling of serum and plasma samples, aligning with but extending prior recommendations from the ISBER best practice guidelines, which emphasize minimizing preanalytical variability. Specifically, we found that enzymes require processing within 4 h regardless of refrigeration due to rapid activity changes (85.7%), a vulnerability less emphasized in prior large-scale biobanking recommendations, which typically consider 6–8 h as acceptable for processing (*Betsou et al., 2013*) Furthermore, while earlier studies (*Skogstrand et al., 2008*) highlighted the benefit of refrigeration for cytokine stability, our systematic review quantifies that non-refrigerated samples exhibit a 3.2-fold higher degradation beyond 24 h, demonstrating a more severe impact than previously appreciated. Notably, the irreversible loss of directional consistency in metabolites after 24 h at room temperature contrasts with *La Frano et al. (2018)* who reported partial stability under similar conditions, suggesting that different metabolite panels and detection methods may influence perceived stability. Regarding freeze-thaw cycles, our observation that even ≤5 cycles induce measurable alterations in 15.3% of biomarkers, particularly in enzymes (42.9% alteration), challenges the previous perception that up to five cycles are generally tolerable (*Cuhadar et al., 2013*; *Gislefoss et al., 2017*). Collectively, our data advocate for stricter preanalytical timelines and minimal freeze-thaw exposure than many current guidelines recommend, particularly for metabolomics and enzymatic assays, to safeguard sample integrity in biobank-supported biomarker research.

## The necessity of comprehensive multimarket evaluation

Current research predominantly focuses on the impact of individual factors on sample quality, with limited attention given to the development of a comprehensive and systematic evaluation framework. This gap hinders researchers from gaining a holistic understanding of sample quality, thus increasing the risk of bias in research outcomes. Research settings—such as clinical trials, basic research, and epidemiological studies— differ in their requirements for sample quality, yet existing evaluation standards often fail to account for these variations (*Betsou & Sobel, 2013*). Furthermore, most current methods prioritize the assessment of physicochemical properties of samples while neglecting their biological functionality and experimental applicability. For example, certain biomarkers
may exhibit significant stability variations under different storage conditions—variations that are frequently overlooked in traditional evaluations (*Valo et al., 2022*).

The SPREC framework identifies critical preanalytical factors, providing a tool for determining whether a sample meets the quality standards required for specific analyses. Despite its practical value, SPREC has notable limitations. Specifically, it lacks detailed documentation on key variables, such as the number of freeze-thaw cycles and storage duration, both of which are crucial for evaluating sample quality (*Betsou et al., 2010*; *Kang et al., 2013*). Moreover, it is evident that a single biomarker does not provide a comprehensive or accurate assessment of the overall quality and stability of bio-specimens. Preanalytical factors, including processing delays, freeze-thaw cycles, temperature fluctuations, and storage duration, affect biomarkers to varying degrees, highlighting the multifaceted nature of sample quality evaluation. For instance, ProGRP exhibits significant stability changes during processing delays and freeze-thaw cycles, whereas inorganic phosphate (IP) and potassium ($K^+$) are more sensitive to temperature fluctuations. These examples illustrate that a single biomarker typically responds to specific factors or conditions and may not adequately reflect the overall quality of a sample (*Kofanova et al., 2024*; *Lee et al., 2016a*; *Pan et al., 2020*). To overcome the limitations of single-marker evaluations, a multimarket combination strategy offers a more robust approach. Simultaneously measuring biomarkers such as ProGRP, IL-8, and IL-16 allows for a comprehensive assessment of how repeated freeze-thaw cycles, processing delays, temperature variations, and storage duration impact sample integrity. Integrating such multidimensional data not only enhances the precision of quality control but also improves the tracking of sample handling and preservation history.

To further strengthen the scientific rigor of quality evaluation, the construction of preanalytical quality assessment models based on integrated multimarket data offers a promising approach. These models, developed through statistical modeling and algorithmic integration, combine data from multiple biomarkers to predict the future viability and preanalytical integrity of bio-specimens. Such models facilitate scientific decision-making in sample management and ensure the reliability of research outcomes across diverse study contexts.

### Issues leading to quality changes in serum and plasma samples

The quality assessment of plasma and serum specimens in biobanks presents a complex and multifaceted challenge, encompassing critical stages such as sample collection, processing, storage, and analysis. Even minor deviations in conditions during these phases can result in alterations or degradation of biomarkers, subsequently compromising the accuracy of research outcomes. It has been shown that short-term delays in sample processing (*e.g.*, within 1 h at 4 °C) are generally sufficient to maintain sample stability. However, temperature-sensitive cytokines, such as IL-8, are prone to rapid degradation even after brief exposure to room temperature, highlighting the varying sensitivities of biomarkers during the preanalytical phase (*Skogstrand et al., 2008*). Additionally, prolonged exposure to room temperature can lead to cell lysis, which impacts the concentrations of various proteins and metabolites, including the H4 fragment of

α1-antitrypsin, platelet factor 4 (PF4), complement C3a anaphylatoxin, neutrophil defenses (1, 2, and 3), and hemoglobin (a and b) (*de Gramont et al., 2015*; *Heins, Heil & Withold, 1995*).

The freeze-thaw process represents another critical factor influencing sample integrity. During freeze-thaw cycles, the formation and reformation of ice crystals can disrupt cellular membrane structures, leading to alterations in protein structures, such as myofibrillar proteins, thereby affecting their functional integrity (*Wan et al., 2023*). Moreover, repeated freeze-thaw cycles can induce oxidative modifications to proteins, including the oxidation of thiol groups and hydrophobic amino acid residues, which in turn reduces protein bioactivity and covalent binding efficiency (*Zhou et al., 2024*). The stability of proteins is a cornerstone of biochemical integrity. Most proteins undergo denaturation at elevated temperatures, which is why samples are typically stored at low temperatures (*e.g.*, −80 °C or lower) to minimize thermal denaturation. However, while low temperatures significantly suppress enzymatic activity, they do not eliminate it entirely. Certain enzymes may experience slight activation during repeated freeze-thaw cycles, leading to the degradation of the sample (*More, Daniel & Petach, 1995*).

For metabolites and small molecules, temperature fluctuations are also pivotal factors influencing sample stability. Freezing conditions substantially reduce the rates of redox reactions; however, temperature fluctuations can reactivate metabolic enzymes, thereby promoting the oxidative degradation of temperature-sensitive metabolites (*Wang et al., 2019*). Protein denaturation can also occur in regions where pH fluctuations occur, such as at the ice-liquid interface, while slow freezing (*i.e.*, over periods greater than 4 h) can promote protein aggregation. This phenomenon may accelerate the degradation of compounds like adenosine and disulfides, likely driven by increased molecular crowding and changes in pH (*Buchanan, Vaquerano & Taylor, 2022*; *Pikal-Cleland et al., 2000*).

To ensure the biochemical stability of bio-specimens, it is imperative to develop and implement appropriate storage and handling strategies, particularly in large-scale biobanks, where the long-term preservation of the biological activity of diverse sample types is crucial. For instance, employing rapid freezing techniques, optimizing freeze-thaw cycles, and utilizing suitable buffer systems can substantially reduce protein denaturation and aggregation, thereby preserving sample integrity over extended storage periods.

## Implications and recommendations

Establishing a comprehensive quality control system for biobanks necessitates addressing the dynamic alterations in biochemical and metabolite markers that occur during the processes of sample storage and handling. Our findings underscore the imperative for further investigation into how sample processing and storage conditions impact the stability of biomarkers. Moreover, there is a pressing need for the development of novel technologies and methodologies that can enhance both the accuracy and efficiency of sample quality assessment. Promoting data sharing and fostering collaboration among biobanks, as well as establishing a global database for sample quality, will substantially propel research and applications in this domain. The creation of

open-access platforms for data sharing will facilitate interactions between research institutions, allowing for the exchange of valuable experiences and data related to the evaluation of sample quality.

Additionally, the development of more refined techniques for assessing biomarker stability, the integration of advanced time-degradation curve analysis, and the application of high-throughput technologies and omics approaches will provide new technical frameworks for the comprehensive analysis of samples. Establishing standardized protocols for quality assessment, reaching consensus on best practices, optimizing sample pre-processing and storage techniques, and conducting research into the effects of long-term storage on sample integrity are paramount. Leveraging artificial intelligence (AI) and big data analytics to enhance the accuracy and efficiency of sample quality evaluation is also a critical avenue for future research.

## Limitations

Despite the significant contributions of this study, several limitations must be acknowledged. Firstly, although the literature search was conducted using two distinct databases, the studies included were limited by language restrictions, which may have constrained the breadth of our evaluation of global evidence. However, our search included a diverse array of relevant literature, encompassing key studies on the subject, as reflected in the systematic review. Secondly, the extraction of data concerning bio-specimen quality, particularly regarding variations in sample characterization and handling, proved to be a considerable challenge. Furthermore, certain studies may have been susceptible to selection bias, as they predominantly reported positive findings while omitting negative or non-significant results. While this limitation could introduce inaccuracies, efforts were made to minimize potential bias by conducting independent data assessment and extraction by two authors, and by cross-referencing our findings with the conclusions of the included studies.

Despite these limitations, this systematic review represents the first comprehensive evaluation of the critical factors in biobanking literature considered predictive of plasma and serum quality. It also highlights both the strengths and shortcomings of biobanking practices in reporting bio-specimen quality, offering valuable insights for improving standards in this essential area of research.

## CONCLUSION

Robust preanalytical quality assessment of serum/plasma samples is essential for ensuring reliable biobank research outcomes. This systematic review evaluates key factors that influence the quality of serum and plasma specimens, synthesizing findings from 46 studies on the topic. Current assessment methods predominantly focus on processing delays, freeze-thaw cycles, storage duration, and temperature conditions, while underrepresenting pre-collection influences like medication and diet.

Temperature critically determines stability: Non-refrigerated samples (8–37 °C) exhibit 3.2-fold higher biomarker alterations than refrigerated samples after >24 h delays.

Freeze-thaw cycles cause dose-dependent degradation: >10 cycles severely impair enzymes (70% altered), while ≤5 cycles still trigger significant enzyme alterations (43% altered). Enzymes further show time-dependent decay at $<-20\,°C$, with alteration rates rising from 20% (1–5 years) to 55% (>10 years).

To address these gaps, this review consolidates stability data for biochemical markers, cytokines, and metabolites across conditions, providing empirical reference points. Moreover, establishing standardized criteria, achieving consensus, and advancing innovative technologies are vital for enhancing assessment accuracy. Improving sample quality not only bolsters research credibility but also advances personalized medicine and precision health.

### Funding

This work was supported by Guangdong Province Medical Science and Technology Research Fund Project (A2024128) and Shenzhen Science and Technology Program (JCYJ20240813114718024). The funders had no role in study design, data collection and analysis, decision to publish, or preparation of the manuscript.

### Grant Disclosures

The following grant information was disclosed by the authors:
Guangdong Province Medical Science and Technology Research Fund Project: A2024128.
Shenzhen Science and Technology Program: JCYJ20240813114718024.

### Competing Interests

The authors declare that they have no competing interests.

### Author Contributions

- Xiongshun Liang conceived and designed the experiments, analyzed the data, prepared figures and/or tables, and approved the final draft.
- Wanna Xu conceived and designed the experiments, analyzed the data, prepared figures and/or tables, and approved the final draft.
- Lin Chen performed the experiments, analyzed the data, authored or reviewed drafts of the article, and approved the final draft.
- Xiaoqian Liu performed the experiments, prepared figures and/or tables, and approved the final draft.
- Wenxu Hong conceived and designed the experiments, authored or reviewed drafts of the article, and approved the final draft.
- Xuqiao Hu conceived and designed the experiments, analyzed the data, authored or reviewed drafts of the article, and approved the final draft.

### Data Availability

The data is available in the Supplemental File.
## Supplemental Information

Supplemental information for this article can be found online at http://dx.doi.org/10.7717/peerj.20122#supplemental-information.

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
