# Peer review of "Quality assessment from biobank plasma and serum specimens: a systematic review"

_PeerJ, doi:10.7717/peerj.20122_

## Round 0.1 · original submission · Major Revisions

Please address the concerns raised by the reviewers and amend the manuscript accordingly.

Reviewer 1 ·

Basic reporting

In this review, the authors report on the importance of the high quality of biobanked samples. The focus of the plasma and serum samples. The molecular composition of human blood derivatives can provide valuable insights into biomedical studies. Thus, the reliability of the results of these studies is of crucial importance and is strictly related to the quality of the samples used for the analysis. Thus, the topic of the review is of high importance.
Regarding point 4 at line 79 - “what methods and technologies are currently employed to evaluate the quality of serum and plasma samples?” – the use of NMR is missing and should be mentioned. In fact, in the last decades, NMR has been extensively used to evaluate the quality of biological samples, especially for those dedicated to metabolomics studies. Moreover, regarding the field of metabolomics, several studies (based both on NMR and MS technologies) have been performed to characterize and minimize the degradation events occurring during the pre-analytical phase in biological samples dedicated to metabolomic studies; importantly, they have led to the development of ISO standards for metabolomics analysis of blood and urine samples “ISO 23118:2021- “Molecular in vitro diagnostic examinations – specifications for pre-examination processes in metabolomics in urine, venous blood serum and plasma”. As specified in the abstract of the ISO document, this document specifies requirements and gives recommendations for the handling, documentation, and processing of urine, venous blood plasma, and serum intended for metabolomics analysis in the pre-examination processes. This document is applicable to metabolomics examinations and can be used by biomedical laboratories, customers of laboratories, in vitro diagnostics developers and manufacturers, institutions and companies performing biomedical research, biobanks, and regulatory authorities.
See, for example:

-N Biotechnol. 2019 Sep 25;52:25-34. doi: 10.1016/j.nbt.2019.04.004.
-Anal Chim Acta. 2021 Oct 16;1182:338968. doi: 10.1016/j.aca.2021.338968.
-N Biotechnol. 2022 May 25;68:37-47. doi: 10.1016/j.nbt.2022.01.006.
-Clin Chem. 2014 Feb;60(2):399-412. doi: 10.1373/clinchem.2013.211979.
-Clin Chem. 2021 Aug 5;67(8):1153-1155. doi: 10.1093/clinchem/hvab092.
- B. Kamlage, S. Neuber, B. Bethan, S. González Maldonado, A. Wagner-Golbs, E. Peter, et al.
- J Biomol NMR. 2011 Apr;49(3-4):231-43. doi: 10.1007/s10858-011-9489-1.
- Metabolites. 2018 Jan 13;8(1):6. doi: 10.3390/metabo8010006.
In my opinion, it is important that these topics are mentioned in the text of this review.

Experimental design

Some important papers are missing (see comment above).

Validity of the findings

no comment

·

Basic reporting

The manuscript text is mostly clear, but it would benefit from reviewing/editing but someone for whom English is their first language. There are sections where many words are inappropriately hyphenated, possibly due to being copied from another space-constrained format.
Also, the Research Trends section repeatedly uses "and so on" after lists. This is not typically used in academic writing.
There are several sections/sentences that are exact duplicates in the text. For example, lines 170- 172 and 173-174, 358-363 and 363-368; the latter of these duplications cites a different reference to support the same statement.
It is not typical to include cited authors' first names (e.g., lines 396 and 399).

Experimental design

The databases searched are not explicitly stated. Figure 1 indicates PubMed and Web of [Sic]

Validity of the findings

This systematic review does not present the results of the reviewed literature in a usefully informative way. The figures present the number of papers published by year, by journal, by country, by topic, and by assessment technique. However, the utility of such a review would be in presenting the quantitative assessments of the impact of the aspects of storage and processing that are reviewed. Unfortunately, the authors merely state very qualitative conclusions for each factor, such a, lines 208-214:

"While short-term storage has a relatively minor effect on the stability of certain cytokines, extended storage may alter cytokine concentrations, emphasizing the need for careful monitoring of these indicators in biobanks(van Waateringe et al. 2017). Long-term storage at -80°C or lower temperatures is considerably more effective in preserving the stability of various biochemical indicators. For instance, hsCRP maintains stability for over four years when stored at -80°C(van Waateringe et al. 2017). Similarly, long-term storage exerts minimal effects on cytokines; multiple cytokines, including IL-8 and IL-16, remain stable under -80°C conditions(Kofanova et al. 2024)."

These qualitative statements, such as "Relatively minor effect [on stability]", "may alter cytokine concentrations", are not useful beyond steering the reader towards potentially relevant references. A more informative approach would be to tabulate the quantitative results presented in the original literature so that the reader could assess effect sizes associated with specific storage and processing attributes.

---

## Round 0.2 · Minor Revisions

Please address the remaining issues of Reviewer #2 and amend the manuscript accordingly.

Reviewer 1 ·

Basic reporting

The revised manuscript is suitable for publication

Experimental design

The revised manuscript is suitable for publication

Validity of the findings

The revised manuscript is suitable for publication

Additional comments

The revised manuscript is suitable for publication

·

Basic reporting

As mentioned in the original review, there are still examples of hyphenated words that do not need to be hyphenated. It seems that these were introduced from previous formatting. Recommend doing search for hyphens and eliminating unnecessary inclusions.

Experimental design

The authors have addressed original concerns regarding inadequate description of search strategy.
Recommend adding PRISMA reference in Methods section (line 78) where they state that they followed "established methodological standards".

Validity of the findings

The resubmission is greatly improved by including tables that quantify the frequency of significant deviations in assay results according to pre-analytical factors.

Introduction, line33-34: Implies that a centralized "United States Biobank has accumulated over 300 million samples"; this number is an estimate for the whole United States, not a single entity.

Additional comments

In the abstract (lines 24-25), the authors state "Standardizing procedures to address
these gaps will enhance sample integrity, research reliability, and clinical utility." However, much of the research conducted is with samples that have already been collected/stored. Surely a key message of this review is to acknowledge and understand the potential limitations of existing biospecimen collections, particularly for certain types of assays.

---

## Round 0.3 · accepted · Accept

All remaining concerns of the reviewer were adequately addressed and the revised manuscript is acceptable now.